https://doi.org/10.1038/s41467-022-29440-z | **OPEN**

# Neuropathology and virus in brain of SARS-CoV-2 infected non-human primates

Ibolya Rutkai[1,8], Meredith G. Mayer[2,8], Linh M. Hellmers[2], Bo Ning[3], Zhen Huang [3], Christopher J. Monjure[2], Carol Coyne[2], Rachel Silvestri[2], Nadia Golden[2], Krystle Hensley[2], Kristin Chandler[2], Gabrielle Lehmicke[2], Gregory J. Bix [4], Nicholas J. Maness [2,5], Kasi Russell-Lodrigue [2,6], Tony Y. Hu [3], Chad J. Roy [2,5], Robert V. Blair [2,7], Rudolf Bohm[2,6], Lara A. Doyle-Meyers[2,6], Jay Rappaport[2,5] & Tracy Fischer [2,5✉]

Neurological manifestations are a significant complication of coronavirus disease (COVID-19), but underlying mechanisms aren't well understood. The development of animal models that recapitulate the neuropathological findings of autopsied brain tissue from patients who died from severe acute respiratory syndrome coronavirus 2 (SARS-CoV-2) infection are critical for elucidating the neuropathogenesis of infection and disease. Here, we show neuroinflammation, microhemorrhages, brain hypoxia, and neuropathology that is consistent with hypoxic-ischemic injury in SARS-CoV-2 infected non-human primates (NHPs), including evidence of neuron degeneration and apoptosis. Importantly, this is seen among infected animals that do not develop severe respiratory disease, which may provide insight into neurological symptoms associated with "long COVID". Sparse virus is detected in brain endothelial cells but does not associate with the severity of central nervous system (CNS) injury. We anticipate our findings will advance our current understanding of the neuropathogenesis of SARS-CoV-2 infection and demonstrate SARS-CoV-2 infected NHPs are a highly relevant animal model for investigating COVID-19 neuropathogenesis among human subjects.

[1] Department of Pharmacology, Tulane University School of Medicine, New Orleans, LA, USA. [2] Tulane National Primate Research Center, Covington, LA, USA. [3] Department of Biochemistry and Molecular Biology, Center for Cellular and Molecular Diagnostics, Tulane University School of Medicine, New Orleans, LA, USA. [4] Department of Neurosurgery, Clinical Neuroscience Research Center, Tulane University School of Medicine, New Orleans, LA, USA. [5] Department of Microbiology and Immunology, Tulane University School of Medicine, New Orleans, LA, USA. [6] Department of Medicine, Tulane University School of Medicine, New Orleans, LA, USA. [7] Department of Pathology and Laboratory Medicine, Tulane University School of Medicine, New Orleans, LA, USA. [8] These authors contributed equally: Ibolya Rutkai, Meredith G. Mayer. ✉email: tfischer1@tulane.edu

Multiple and continuing reports demonstrate a substantial number of patients with coronavirus disease 2019 (COVID-19) develop new-onset neurological symptoms. Several case reports have, in fact, identified neurological complications as the initial presentation of severe acute respiratory syndrome coronavirus 2 (SARS-CoV-2) infection, particularly among those who develop stroke[1–3]. Among the more urgent COVID-19-associated neurological presentations, stroke, meningoencephalitis, and hemorrhagic necrotizing encephalopathies have been associated with more severe disease[2,4–6]; however, even comparatively mild neurological symptoms, such as dizziness or unresolving headache[4,7], may be indicative of neuropathological processes in the context of infection and disease. Notably, individuals across the lifespan, with and without significant comorbidities, and with all disease severities, including asymptomatic patients, have suffered the variety of reported neurological manifestations[8].

While damage to the central nervous system (CNS) of patients with COVID-19 is increasingly evident, the neuropathogenesis remains unclear. Here, we provide a comprehensive assessment of brain pathology associated with SARS-CoV-2 infection in two non-human primate (NHP) models of infection with varied disease severity. This work reveals neuroinflammation, brain hypoxia, microhemorrhages, and pathology consistent with hypoxic-ischemic injury with rare infection of brain vasculature in SARS-CoV-2 infected NHPs and provides key insights into SARS-CoV-2-associated neuropathogenesis. Our findings are consistent with those reported on autopsied brain of human subjects who died with SARS-CoV-2 infection. Additional molecular analyses on brain from our animal models suggest reduced oxygen to the CNS may contribute significantly to injury in the context of infection. Importantly, animals that did not develop acute respiratory distress syndrome (ARDS) demonstrated neuropathology that may lead to long-term neurological symptoms of post-acute sequelae of COVID-19 (PASC), or "long COVID".

## Results

**Significant inflammation in brain**. Eight adult NHPs, including four Rhesus macaques (RM), 13–15 years of age, and four wild-caught African green monkeys (AGMs), approximately 16 years of age, were inoculated with the 2019-nCoV/USA-WA1/2020 strain of SARS-CoV-2[9] via a multi-route mucosal or aerosol challenge (Table 1). Two animals of each species were inoculated via aerosol and two by multi-route exposure. Multi-route mucosal exposure included conjunctival, nasal, pharyngeal, and intratracheal routes. Control animals included two RMs, approximately 18–22 years of age, and two AGMs, approximately 17 years of age. Control animals were mock-infected through multi-route mucosal exposure of the same growth media used for virus propagation. All study animals underwent the same clinical tests and procedures.

All animals exposed to SARS-CoV-2 developed infection within the first week of exposure, as demonstrated by the detection of the viral nucleocapsid (N) mRNA in nasal swabs taken within 3–7 days after challenge (Table 1). No differences in infection were noted between the two inoculation strategies. Further verification of infection is seen through detection of the virus by immunohistochemistry (IHC) in lung (Supplementary Data Fig. 1). Additional detailed findings in lung and clinical measures have been previously reported[10].

All animals survived to study endpoint, except for AGM1 and AGM2. At 8 days post infection, AGM1 was found recumbent and marginally responsive to stimuli. This animal also presented with dyspnea/tachypnea (respiratory rate of 72 breaths per minute), hypothermia (<32.2 °C), and hypoxemia [blood oxygen saturation (SpO$_2$) = 77%] and was euthanized. At 22 days post infection, shortly before its scheduled study endpoint, AGM2 developed severe tachypnea, hypothermia, and hypoxemia, with a respiratory rate of 96 breaths per minute and SpO$_2$ = 77% and was subsequently euthanized.

Seven regions of the CNS, including frontal, parietal, occipital, and temporal lobes, basal ganglia, cerebellum, and brainstem were collected at necropsy from all animals and investigated for neuroinflammation through histopathological and immunohistochemical methods. A summary of the neuropathological findings is included in Table 2.

Neuroinflammation was seen in all study animals but was greater in those with SARS-CoV-2, as compared to age-matched mock-infected controls (Fig. 1). The pan-microglial protein, ionized calcium-binding adapter molecule 1 (Iba-1), was upregulated in the context of infection and revealed morphological alterations indicative of microglial activation, with retracted,

---

**Table 1 Study animals.**

| Animal ID | Age (years) | Sex | Species | Route of challenge | Virus exposure | SARS-CoV-2 N mRNA (Eq. VC/mL) | Necropsy (days PI) |
|---|---|---|---|---|---|---|---|
| RM1 | 14.01 | Male | *M. mulatta* | Multi-route | $3.61 \times 10^6$ PFU | $1.46 \times 10^4$ | 27 |
| RM2 | 12.97 | Female | *M. mulatta* | Multi-route | $3.61 \times 10^6$ PFU | $2.86 \times 10^6$ | 27 |
| RM3 | 13.06 | Male | *M. mulatta* | Aerosol | $2 \times 10^3$ TCID$_{50}$[a] | $2.97 \times 10^7$ | 28 |
| RM4 | 15.03 | Male | *M. mulatta* | Aerosol | $2 \times 10^3$ TCID$_{50}$[a] | $1.29 \times 10^9$ | 28 |
| RM5 | 17.97 | Female | *M. mulatta* | Multi-route | TC media | n/a | 29 |
| RM6 | 21.62 | Male | *M. mulatta* | Multi-route | TC media | n/a | 29 |
| AGM1 | 16.28 | Female | *C.a. sabaeus* | Aerosol | $2 \times 10^3$ TCID$_{50}$[a] | $9.33 \times 10^4$ | 8 |
| AGM2 | 16.29 | Female | *C.a. sabaeus* | Multi-route | $3.61 \times 10^6$ PFU | $6.94 \times 10^4$ | 22 |
| AGM3 | 16.3 | Male | *C.a. sabaeus* | Multi-route | $3.61 \times 10^6$ PFU | $7.62 \times 10^3$ | 26 |
| AGM4 | 16.33 | Male | *C.a. sabaeus* | Aerosol | $2 \times 10^3$ TCID$_{50}$[a] | $2.58 \times 10^4$ | 24 |
| AGM5 | 17.34 | Female | *C.a. sabaeus* | Multi-route | TC media | n/a | 28 |
| AGM6 | 17.34 | Male | *C.a. sabaeus* | Multi-route | TC media | n/a | 28 |

Four Rhesus macaques and four African green monkeys were exposed to SARS-CoV-2 strain 2019-nCoV/USA-WA1/2020 by multiple mucosal routes or aerosolized virus in an aerosol chamber. All infected animals survived to the study endpoint at 24–28 days post infection, with the exception of AGM1 and AGM2 who reached humane endpoints at 8- and 22 days post infection, respectively. The establishment of infection was demonstrated by the detection of SARS-CoV-2 nucleocapsid (N) mRNA in nasal swabs within the first week of exposure (3–7 days after challenge). Two Rhesus macaques and two African green monkeys were used as age-matched, mock-infected controls for the study.
*PFU* plaque-forming unit, *TC* tissue culture, *TCID50* 50% tissue culture infectious dose, *M. mulatta* Macaca mulatta (Rhesus macaque, RM), *C.a. sabaeus* Chlorocebus aethiops sabaeus (African green monkey, AGM).
[a]Approximate inhaled dose.

**Table 2 CNS Pathology and Summary of Findings.**

| ID | Route of infection | Age (years) | Sex | CNS pathology and summary of findings |
|---|---|---|---|---|
| RM1 | Multi-route mucosal | 14.01 | Male | Multiple acute microhemorrhages (++++) were observed in cerebellum, BG, and brainstem. Marked neuronal and surrounding cell injury was seen within cerebellum and brainstem. Cleaved caspase 3 positivity was mostly limited to Purkinje cells and immediate neighbors (+++) in cerebellum. BG had a single region of cell with sporadic caspase 3 positivity within the area. Limited vascular caspase 3 positivity was also observed in the BG (+). No caspase 3 positivity was observed in brainstem despite large regions with abnormal neuronal morphology. Limited SARS-N positivity was found in endothelium of cerebellum, brainstem, and BG (+). |
| RM2 | Multi-route mucosal | 12.97 | Female | Acute microhemorrhages were seen in cerebellum and BG (+). Marked neuronal injury was present in cerebellum and brainstem. Cleaved caspase 3 positivity was observed in Purkinje cells and immediate neighbors in cerebellum and parenchymal cells in brainstem (+++). Rare caspase 3 positivity was observed in cerebellar endothelium, with much greater EC positivity seen in brainstem (+++). Caspase 3 was also observed in BG endothelium, but to a lesser degree (+). BG also showed limited caspase 3 positivity of parenchymal cells with apparent nuclear dissolution and surface blebs (+). Parietal lobe showed rare cleaved caspase 3 positivity in ECs and parenchymal cells. This was localized to blood vessels within associated areas of tissue damage that contained cells at different stages of nuclear dissolution with apparent blebbing. The temporal lobe had several foci with high cleaved caspase 3 positivity (+++). Cleaved caspase 3 was also seen with moderate frequency in temporal lobe ECs (++). Limited SARS-N positivity in endothelium of cerebellum and brainstem (+), with infrequent positivity observed in BG. |
| RM3 | Aerosol | 13.06 | Male | An acute microhemorrhage was seen in the BG but not in cerebellum or brainstem, in contrast to the majority of our study animals. Moderate neuronal injury with vacuoles in WM were seen in cerebellum. Rare cleaved caspase 3 positivity was seen in Purkinje cells and immediate neighbors. Limited pyknotic neurons were observed in brainstem, however, infrequent caspase 3 positivity was restricted to the endothelium. Rare cleaved caspase 3 positivity was also observed in parietal lobe, despite apparent areas of cell injury/death. Rare SARS-N positivity was detected in endothelium of brainstem, BG, and parietal and temporal lobes. |
| RM4 | Aerosol | 15.03 | Male | A moderate number of acute microhemorrhages (++) were seen in cerebellum, BG, and brainstem. Marked neuronal injury was observed in cerebellum with WM vacuolation. Active caspase 3 positivity was not detected. In brainstem, foci of cell injury/apoptosis were seen with active caspase 3 positivity in parenchymal cells and endothelium (++). Rare SARS-N positivity was found in endothelium of cerebellum and temporal lobe, with infrequent positivity in parietal and occipital lobes. |
| RM5 | (mock) Multi-route mucosal | 17.97 | Female | Acute areas of neuronal injury in the Purkinje cell layer of the cerebellum (+). No microhemorrhages were seen in any regions. All brain regions were negative for cleaved caspase 3 and SARS nucleocapsid. |
| RM6 | (mock) Multi-route mucosal | 21.62 | Male | Healthy brain morphology was generally observed. A microhemorrhage was noted in the brainstem, basal ganglia, and cerebellum. All brain regions were negative for cleaved caspase 3 and SARS nucleocapsid. |
| AGM1 | Aerosol | 16.28 | Female | Extensive acute microhemorrhages (+++) were also seen in cerebellum, BG, and brainstem. Marked neuronal and neighboring cell injury were also seen in cerebellum, BG, and brainstem but without cleaved caspase 3 positivity. Likewise, cleaved caspase 3 was not seen in parietal lobe, despite obvious cell/tissue injury and/or death. Rare SARS-N positivity was detected in endothelium of cerebellum, brainstem, and BG, which was infrequent and dim in the temporal lobe. |
| AGM2 | Multi-route mucosal | 16.29 | Female | A considerable number of acute microhemorrhages (++++) were observed in cerebellum, BG, and brainstem. While marked neuronal injury was observed in cerebellum, cleaved caspase 3 positivity in Purkinje cells and immediate neighbors was moderate (++). Brainstem had foci of caspase 3 positivity (++), whereas the parietal lobe contained regions of apparent cell injury/death without cleaved caspase 3 positivity. Rare SARS-N positivity was noted in endothelium of cerebellum, brainstem, BG, and parietal lobe. |
| AGM3 | Multi-route mucosal | 16.3 | Male | Several acute microhemorrhages (+++) were seen in cerebellum, BG, and brainstem. Marked neuronal caspase 3 positivity was seen in Purkinje cells and immediate neighboring cells (+++) within the cerebellum, whereas EC-associated positivity was rare. Considerable caspase 3 positivity was present in parenchymal and ECs of brainstem (++++) and parietal lobe [parenchymal (+++); ECs (+)]. Rare SARS-N positivity was observed in the endothelium of cerebellum, brainstem, BG, and temporal lobe. |
| AGM4 | Aerosol | 16.33 | Male | Several acute microhemorrhages (+++) were seen in cerebellum and brainstem. Cerebellum showed marked neuronal injury with moderate cleaved caspase 3 positivity in Purkinje cells and immediate neighbors (++). Rare caspase 3 positivity was seen in cerebellar and BG endothelial cells. Focal regions of parenchymal cell injury were present in brainstem with active caspase 3 positivity in parenchymal cells and ECs (+). A single region within the parietal lobe had considerable cleaved caspase 3 positivity in the parenchyma (++++ for this area only). Limited SARS-N positivity was seen in endothelium of cerebellum, brainstem, and BG (+), which was infrequent in temporal, parietal, and occipital lobe endothelium. |
| AGM5 | (mock) Multi-route mucosal | 17.34 | Female | Several microhemorrhages seen in the cerebellar white matter and granular layer (+). Rare cleaved caspase 3 positivity in the frontal lobe. All brain regions were negative for SARS nucleocapsid. |
| AGM6 | (mock) Multi-route mucosal | 17.34 | Male | Microhemorrhages were noted in the basal ganglia, brainstem, and cerebellum with larger bleeds in the white matter (+). Rare vacuoles in cerebellum (+). Sparse cleaved caspase 3 positivity in the cerebellum, basal ganglia, and brainstem but not near the areas of microhemorrhages. All investigated brain regions were negative for SARS nucleocapsid. |

A within laboratory scoring scale, ranging from limited (+), mild (++), moderate (+++), and severe (++++) indicates the degree of positivity of specific antigens investigated or severity of observed pathology.
BG basal ganglia, WM white matter, ECs endothelial cells, RM Rhesus macaque, AGM African green monkey.

thickened processes and a large cell body (Fig. 1b, d). Occasional, small perivascular cuffs were observed in infected (Fig. 1f, h) but not control animals (Fig. 1e, g). In contrast, nodular lesions were seen more frequently than cuffs and were present in both infected (Fig. 1j, l) and mock-infected (Fig. 1i, k) animals, however, these appeared larger in the context of infection.

To further characterize microglial activation, tissues were investigated for the MHC class II cell surface receptor, HLA-DR (Fig. 1m–p). Similar to findings in brain of aged human subjects[11], microglial expression of HLA-DR was observed in animals without SARS-CoV-2 infection (Fig. 1m, o). Expression was also seen in brain of infected animals (Fig. 1n, p); however, this did not appear greater than those that were mock-infected. HLA-DR did highlight nodular lesions in all animals, which were larger in infection, as seen with Iba-1.

Additional evidence of increased neuroinflammation in infection was seen through glial fibrillary acidic protein (GFAP) IHC, which was upregulated in infected animals (Fig. 1r, t), as compared to age-matched controls (Fig. 1q, s). GFAP immunopositivity revealed astrocytic hypertrophy in the context of aging, suggestive of astrocyte activation, however, this was more pronounced in infection, which also displayed significant loss of individual astrocytic domains.

**Neuronal injury and apoptosis.** Hematoxylin and eosin (H&E; Fig. 2) staining revealed marked changes in neuronal morphology, which was most often observed in cerebellum and brainstem (Fig. 2b–d). Neuronal degeneration was characterized by pyknotic and karyorrhectic nuclei with shrunken cytoplasm and vacuolation in the surrounding neuropil (Fig. 2b–d). The cerebellum contained several regions of degenerate Purkinje neurons that exhibited cellular blebs and debris and cytoplasmic vacuoles (Fig. 2b, c). Contiguous with areas of degenerate Purkinje cells, neurons and glia within the molecular and granular layers appeared pyknotic with condensed, basophilic nuclei (Fig. 2b). Similar morphologic changes were noted in glial cells adjacent to apoptotic neurons in the brainstem (Fig. 2d). In both brainstem and cerebellum, neurons are seen at various stages of nuclear dissolution (Fig. 2b–d). Degeneration of Purkinje cells was further confirmed with FluoroJade C (Fig. 2e, f).

Given the prominent morphologic changes noted within Purkinje cells, we sought to identify the mechanisms underlying these degenerative changes by investigating all brain regions for the presence of cleaved caspase 3, the activated form of this key executioner of apoptosis. Cleaved caspase 3 was seen in at least one CNS region from all infected animals except AGM1, which did not have any positive cells (Fig. 2i, Supplementary Data Fig. 2). Three animals, RM1, AGM3, and AGM4 showed positivity in more than one brain region, while RM2 had cleaved caspase 3 positive cells in all regions examined (Fig. 2i; Supplementary Data Fig. 2). In cerebellum, cytoplasmic and nuclear-cleaved caspase 3 was predominantly restricted to cells within and proximal to the Purkinje cell layer (Fig. 2g). Other CNS regions, including brainstem, had foci of cleaved caspase 3 positivity (Fig. 2h). In comparison to infected animals, mock-infected controls showed little-to-no positivity (Fig. 2i; Supplementary Data Fig. 2). Unbiased quantitation revealed a statistically significant difference in cleaved caspase 3 positivity between infected and mock-infected animals in all brain regions investigated (Fig. 2i). When stratified by species, statistical significance was not achieved by Mann–Whitney $U$ Test, which is likely due to the low number of each species (Supplementary Data Fig. 2). Interestingly, cleaved caspase 3 was not detected in any CNS region examined from AGM1, who was euthanized at 8 days post infection due to advanced illness. This may suggest

programmed cell death in the CNS occurs later in the disease process.

While vacuolation was at times observed in the cerebellar gray and white matter (Supplementary Data Fig. 3a, b), significant demyelination was not a major finding in this study. Luxol Fast Blue (LFB) did reveal localized myelin pallor, suggestive of oligodendrocyte injury and/or loss, in the cerebellum of RM3 and occipital lobe of AGM3 (Supplementary Data Fig. 3c, d).

**Brain microhemorrhages.** Microhemorrhages, as suggested by the presence of erythrocyte extravasation into the brain parenchyma, were identified in all study animals and seen with and without ischemic injury of adjacent tissues, characterized by localized/regional tissue pallor (Fig. 3a–f). Although the number of bleeds varied, all animals were observed to have at least one. Infected animals appeared to have larger bleeds than mock-infected controls, with more dense accumulation of red blood cells on the parenchymal side of the blood vessel (Fig. 3, compare a–d with e, f). Quantitation of microhemorrhages was determined on Axio Scan.Z1 (Zeiss) scanned slides and HALO software (Indica Labs, v2.3.2089.70 and v3.1.1076.405) and normalized by tissue area (Fig. 3g). The whole brain showed a higher increase in the number of microbleeds in infection which reached statistical significance in the basal ganglia (Fig. 3h and Supplementary Data Fig. 4).

Accumulation of cerebral microhemorrhages occurs with aging and are seen most frequently in deep brain structures, including brainstem, basal ganglia, and cerebellum[12]. This may be due to age-associated decrease in arterial elasticity and increased blood pressure on brain microvasculature, as well as other risk factors for vascular injury, such as diabetes and dyslipidemia. Vascular injury can promote thrombosis, or blood clot formation within a blood vessel, which may aid in stopping the brain microbleed or, conversely may underlie microhemorrhages and result in more serious brain injury by impeding the flow of blood in the brain, leading to stroke. To assess the potential contribution of thrombosis to microhemorrhage development in SARS-CoV-2 infection, we examined all brain regions for luminal accumulation of the platelet glycoprotein, CD61 (aka, integrin b-3). This revealed multiple blood vessels with aggregated platelets in both infected and mock-infected animals, which were seen with and without associated microbleeds (Fig. 4a–d). Microhemorrhages without CD61 accumulation were also observed (Fig. 4e, f). Quantitation of total brain microhemorrhages with and without associated CD61 positivity revealed a greater frequency without thrombi (CD61 positivity) in the context of infection, apart from AGM5 who had many bleeds without visible thrombi (Fig. 4g, h). These findings suggest that in the context of infection, leakage of blood vessels without vascular damage/injury occurs more frequently.

**Chronic hypoxemia/brain hypoxia.** Microhemorrhages and ischemia appear to play a central role in neuronal injury observed in this study. The brain is a highly metabolic organ with a limited capacity for energy storage. Due to the significant energy demands of the brain and neurons, a prolonged reduction in blood flow and concomitant reduction in oxygen and glucose can be detrimental to neuronal vitality, in addition to the resulting neurotoxicity of erythrocyte breakdown products and inflammation. Of particular interest is the finding that AGM1, who was found recumbent and minimally responsive to stimuli at 8 days post infection, had a substantial number of microbleeds in the cerebellum, basal ganglia, and brainstem (Table 2). These findings suggest AGM1 suffered multiple acute microhemorrhages that may have contributed to her rapid decline. Alternatively, AGM1's

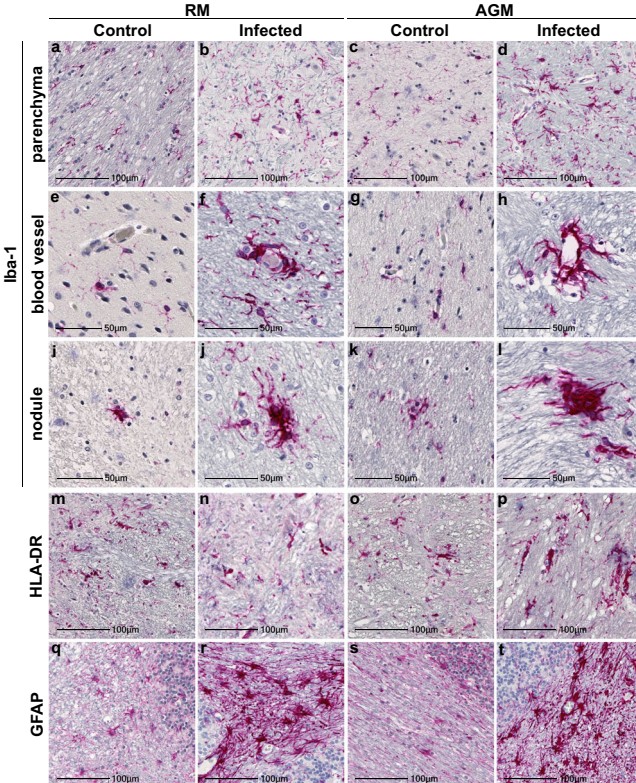

**Fig. 1 Prominent neuroinflammation in brain of SARS-CoV-2 infected NHPs.** Representative images identify microglia through Iba-1 immunopositivity in basal ganglia of mock-infected animals RM6 and AGM5 (**a**, **c**) that was upregulated in SARS-CoV-2 infected parenchyma, as shown in RM2 and AGM4 (**b**, **d**). Mild-moderate accumulation of microglia was often observed around blood vessels (RM1 **f**, AGM1 **h**). Nodular lesions were also frequently observed in brain of infected animals, represented here in RM4 and AGM4 (**j**, **l**). Microglial accumulation around blood vessels was not seen in age-matched mock-infected controls (RM6 **e**, AGM5 **g**), however, nodules (RM5 **i**, AGM5 **k**) were seen. These were less frequent and smaller than those observed in infection. Iba-1 immunopositivity also revealed morphological changes in microglia indicative of increased activation in infected animals, as compared to mock-infected controls, including large cell bodies with short, thickened processes (**b**, **d**, **f**, **h**, **j**, **l**). Microglial expression of HLA-DR was upregulated in the context of infection (**n**, **p**) seen in RM2 and AGM2, however, expression was also seen in control animals (**m**, **o**) represented by RM6 and AGM5. GFAP expression by astrocytes is upregulated and reveals morphological changes in the context of infection (cerebellum from RM4 **r**, AGM2 **t**), indicative of astrogliosis. Cerebellum from non-infected controls RM6 and AGM5 (**q**, **s**). Each immunohistochemical stain was performed twice on all brain regions. Abbreviations: AGM African green monkey, RM Rhesus macaque. Scale bars = 100 μm (**a–d**, **m–t**) and 50 μm (**e–l**).

rapid pulmonary decline may have promoted end stage microhemorrhages. The timing of acute microhemorrhages in the disease process is unclear and warrants further investigation.

In addition to localized ischemic injury, all infected animals experienced variations in $SpO_2$ that fluctuated between 89 and 99% but stayed below 95% for most over the study course (Fig. 5a). Correspondingly, blood carbon dioxide ($CO_2$) ranged from 24 to 33 mEq/L, remaining above the physiological range for most of the study animals (Fig. 5b). While these levels are not immediately alarming, they may suggest mild hypoxemia and impaired gas exchange in the lungs. As such, chronic hypoxemia may contribute to impairment of the endothelium and/or

neurovascular unit leading to increased vascular permeability. The brain requires aerobic metabolism of glucose for ATP production and any prolonged or intermittent reductions of blood $O_2$ may contribute to localized CNS hypoxia and energy failure. Even minor reductions in oxygen may promote injury, particularly among neurons, which appear to have suffered the greatest insult in this study. In support of this notion, large regions of Purkinje cells, which are especially vulnerable to hypoxic insult[13,14], as well as cells in their immediate proximity, appear degenerate or committed to undergoing apoptosis.

To assess brain tissue for evidence of hypoxia, we performed IHC against the oxygen-regulated alpha subunit of hypoxia inducible factor-1 (HIF-1a), which is upregulated and stabilized under hypoxic conditions. For this analysis, only basal ganglia, brainstem, and cerebellum were investigated because our earlier studies demonstrated these brain regions had the greatest injury/pathology. This study demonstrated marked upregulation of HIF-1a in brain of infected animals, as compared to mock-infected controls (Fig. 5f–m). Areas of intense positivity, suggestive of HIF-1a accumulation, were predominantly seen in and around blood vessels, which extended into the brain parenchyma in infection (Fig. 5g, i, k, m). Areas of HIF-1a positivity were noted in mock-infected animals but were less intense than that seen in brain of infected animals and/or did not extend appreciably into the parenchyma (Fig. 5f, h, j, l). Non-biased quantitation of HIF-1a intensity [optical density (OD)] around blood vessels, which excluded the blood vessel lumen, revealed a statistically significant increase in HIF-1a by cells comprising the vasculature and neighboring parenchymal cells of infected animals, as compared to controls, in brainstem (Fig. 5c, *$p = 0.0154$) and basal ganglia (Fig. 5d, **$p = 0.0016$) but not cerebellum (Fig. 5e, $p = 0.0940$). Our approach for quantifying HIF-1a expression around the vasculature, while excluding the blood vessel lumen, is shown in Supplementary Data Figs. 5 and 6. Statistical significance was only retained in the basal ganglia when stratified by species (RMs *$p = 0.049$, AGMs *$p = 0.034$; Supplementary Data Fig. 7).

**Rare virus in brain-associated endothelium.** The potential for direct virus involvement in CNS pathology was explored through IHC and RNAscope analyses of all brain regions. Using an antibody against SARS-CoV-2 nucleocapsid protein (SARS-N), IHC studies revealed rare virus infection in brain that, when seen, appeared to be restricted to the vasculature (Fig. 6a). Sparse virus was detected most frequently within the basal ganglia, cerebellum, and/or brainstem and seen less often within the temporal, parietal, and occipital lobes (Table 2). This was verified further through in situ hybridization (ISH) analyses, employing RNA-scope Technology with enhanced signal amplification. Using an anti-sense probe to the viral spike protein RNA (SARS-S), cytoplasmic positivity was seen in brain of infected animals but not in mock-infected controls (Fig. 6c–h; Supplementary Data Fig. 8). The specificity of the probe used in these studies is demonstrated in lung, which only showed positivity in the context of infection (Supplementary Data Fig. 8).

The single-label studies suggested SARS-CoV-2 infection in brain is limited to the brain vasculature and appeared to be restricted to endothelial cells. Suspected endothelial cell infection is supported by colocalization of SARS-N with von Willebrand factor (vWF; Fig. 6i–k). A blood vessel in close proximity to that shown in Fig. 6i–k but without detectable virus is included to demonstrate the specificity of the SARS-N antibody (Fig. 6l–n).

Using a highly sensitive CRISPR-based fluorescent detection system (CRISPR-FDS)[15], virus was not identified in the cerebrospinal fluid (CSF) (Fig. 6o), consistent with most findings among human subjects, except in rare cases of encephalitis[16–18].

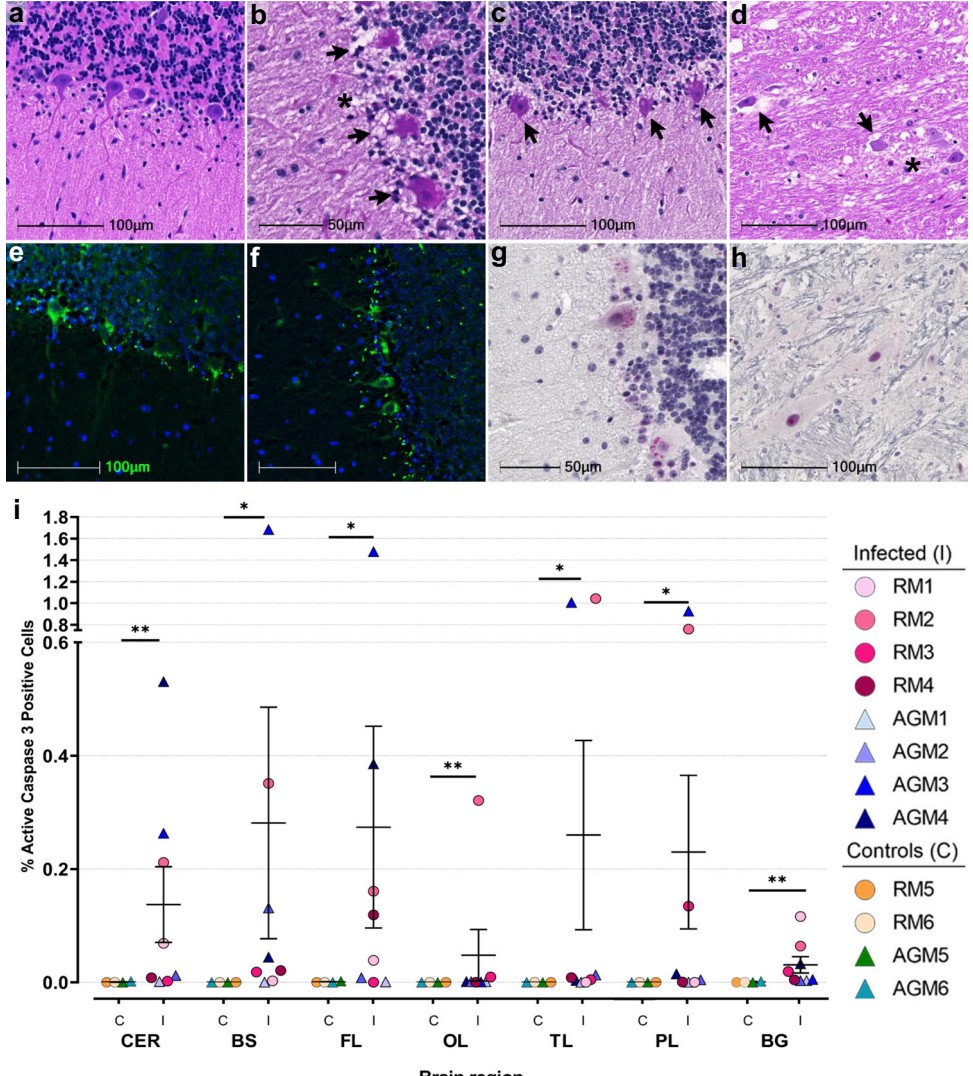

**Fig. 2 Neuronal pathology and cell death in SARS-CoV-2 infection.** Representative H&E images show a healthy Purkinje cell layer in the cerebellum of a non-infected control RM6 (**a**) and reveal cell death-associated neuronal changes in cerebellum of infected animals, (AGM4 **b**, AGM3 **c**), and brainstem from AGM3 (**d**). Arrows indicate pyknotic and karyolitic Purkinje cells and cellular blebs. Asterisks denote areas of tissue necrosis/vacuolation on H&E sections (**b**) and (**d**). H&E was performed and assessed twice on all brain regions. Neuronal degeneration in cerebellum was only seen in the context of infection, visualized by positive, green FluoroJade C-stained neurons (AGM2 **e**, AGM1 **f**). FluoroJade C staining was performed twice on the brain regions investigated. Abnormal neuronal morphology and cleaved caspase 3 positivity is demonstrated in cerebellum (AGM3 **g**) and brainstem (RM2 **h**). Summary data of cleaved caspase 3 positive cells stratified by brain region (**i**) where $n = 4$ biologically independent samples/brain region in the control group and $n = 8$ biologically independent samples/brain region in the infected group, except OL where $n = 7$ infected animals. Immunohistochemical staining for cleaved caspase 3 was performed twice on all brain regions. Statistics were performed with a two-tailed Mann–Whitney $U$ test. *$p \leq 0.05$ and **$p \leq 0.005$ comparing mock-infected to infected animals. Data are expressed as mean ± SEM. $p$ values: CER = 0.0081, BS = 0.0182, FL = 0.0323, OL = 0.0061, TL = 0.0485, PL = 0.0485, BG = 0.0040 (control vs. infected). Source data are provided as a Source Data file. Abbreviations: CER cerebellum, BS brainstem, FL frontal lobe, OL occipital lobe, TL temporal lobe, PL parietal lobe, BG basal ganglia, C control, I infected, AGM African green monkey, RM Rhesus macaque. Scale bars = 100 µm (**a**, **c**–**f**, **h**) and 50 µm (**b**, **g**).

In contrast, this method detected limited viral RNA in whole brain, frozen at the time of necropsy, that was largely representative of our IHC/IF findings (Fig. 6o). Similar to our findings in fixed tissues, virus was more frequently observed in basal ganglia, cerebellum, and brainstem. CRISPR-FDS analysis also revealed viral RNA in the frontal lobe of one animal, AGM1, which was not convincingly seen by IHC/IF for this region in any study animal. This may reflect differences in sampling error that is inherently present in the two methods, where the amount of tissue used for the CRISPR-FDS studies is greater than that used in IHC, and/or extracerebral virus that may have been present in the blood vessel lumen.

Together, our findings demonstrate scarce SARS-CoV-2 infection in brain-associated endothelial cells in deep brain structures of NHPs, even in the absence of severe disease or overt neurological symptoms.

## Discussion

Neurological manifestations are commonly seen in the context of SARS-CoV-2 infection but are highly varied and range in severity from impaired smell and/or taste to stroke[2,19]. As such, the mechanisms underlying SARS-CoV-2-associated neurological complications are likely complex. Relevant animal models of

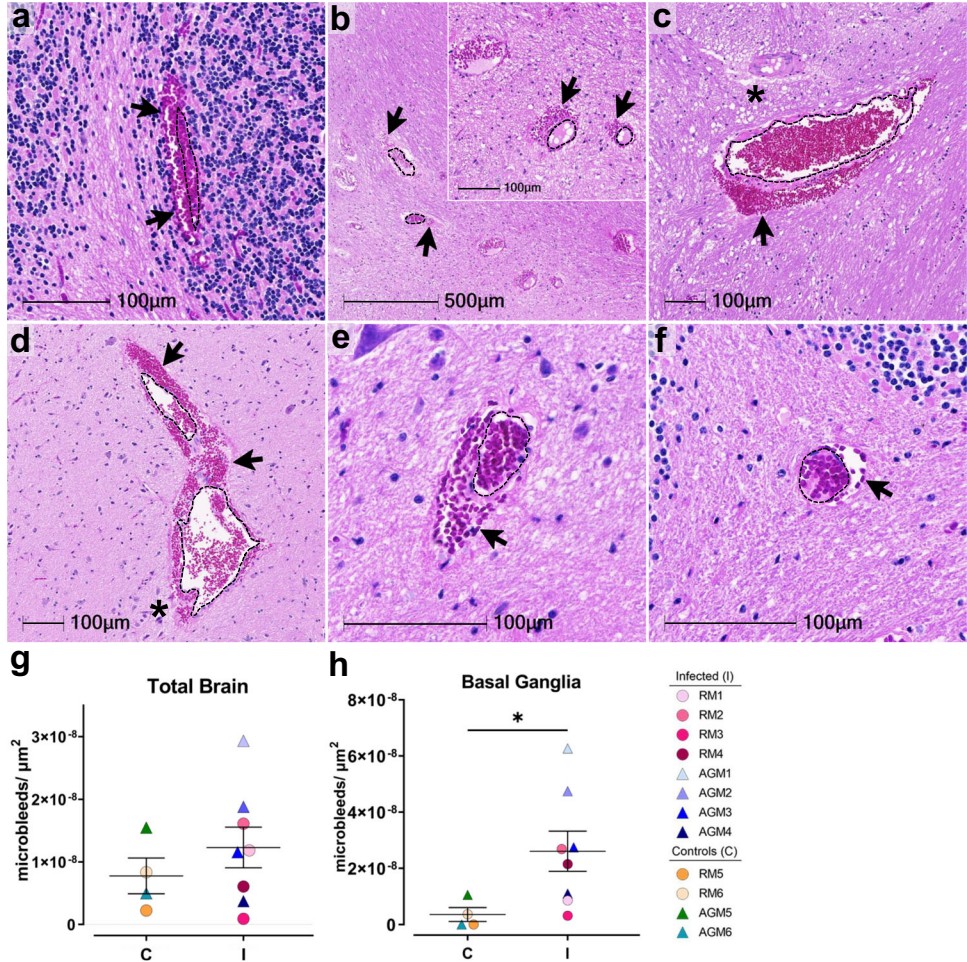

**Fig. 3 Multiple microhemorrhages in CNS of SARS-CoV-2 infected NHPs.** H&E examination of infected animals revealed microhemorrhages, as demonstrated in cerebellum (AGM1 **a**), brainstem (AGM2 **b**, AGM4 **c**), and basal ganglia (RM4 **d**), which tended to be larger and packed with red blood cells, as compared to non-infected controls, (RM6 brainstem **e**, AGM5 cerebellum **f**). Erythrocyte extravasation into the brain parenchyma is indicated by black arrows. Asterisks denote tissue injury around damaged blood vessels. A dotted line outlines the vessel lumen in each panel. H&E was performed and assessed twice on all brain regions. The number of microbleeds/mm$^2$ was assessed in all brain regions (**g**) and found to be significantly greater in the basal ganglia (**h**, *$p$ = 0.0263), where $n$ = 4 biologically independent samples in the control group and $n$ = 8 biologically independent samples in the infected group. Statistics were performed with a two-tailed Mann–Whitney $U$ test. *$p \leq 0.05$. Data are expressed as mean ± SEM. Source data are provided as a Source Data file. Abbreviations: C control, I infected, AGM African green monkey, RM Rhesus macaque. Scale bars = 100 μm except outset of b where scale bar = 500 μm.

infection and CNS involvement that reflect human disease are critical for elucidating these mechanisms, as well as identifying and/or developing effective therapeutic strategies. In our two models of aged NHPs infected with SARS-CoV-2, we found evidence of prominent neuroinflammation, microhemorrhages with and without associated microthrombi, and neuronal injury and death consistent with hypoxic-ischemic injury but without substantial virus detection in brain. Our findings are largely in line with those reported in autopsy studies of individuals who died from infection[20–26]. Like human disease, reactive astrocytes and microglia were a common feature, seen throughout the entirety of the brain in infected animals. This appeared greater in basal ganglia, brainstem, and cerebellum, which contained the majority of cuffs and nodular lesions observed. Lymphocyte infiltrate, which has been reported in human brain[22,24], was not observed in any brain region investigated from our study animals. This may reflect a shorter time with severe disease in our animal model. Additional life-saving efforts were not made for animals that developed serious disease (e.g., ARDS), as would be done with humans, and were quickly euthanized to minimize pain and suffering of the animal. It

is worth noting that autopsy reports of significant lymphocyte infiltration into the CNS or COVID-associated encephalitis are relatively few and may be a less common complication of disease[8].

Our findings of hypoxic-ischemic injury in brain of NHPs are also in agreement with autopsy studies of brain from human subjects[21,27]. This may arise from chronic, peripheral hypoxemia, as well as reduced cerebral blood flow due to acute microhemorrhages. The brain is a highly metabolic organ and requires aerobic metabolism of glucose for adenosine triphosphate (ATP) production. Any prolonged or chronic intermittent reductions of blood SpO$_2$ may contribute to localized CNS hypoxia and energy failure. Even minor, but sustained, reductions in oxygen may promote injury, particularly among neurons, which appear to have suffered the greatest insult in this study. In support of this notion, large stretches of Purkinje cells, which are especially vulnerable to hypoxic insult[13,14], as well as cells within their immediate proximity, appear degenerate and/or committed to undergoing apoptosis. Areas of injured neurons at various stages of nuclear dissolution were noted in other brain regions, including brainstem. Moreover, neuronal injury did not appear to

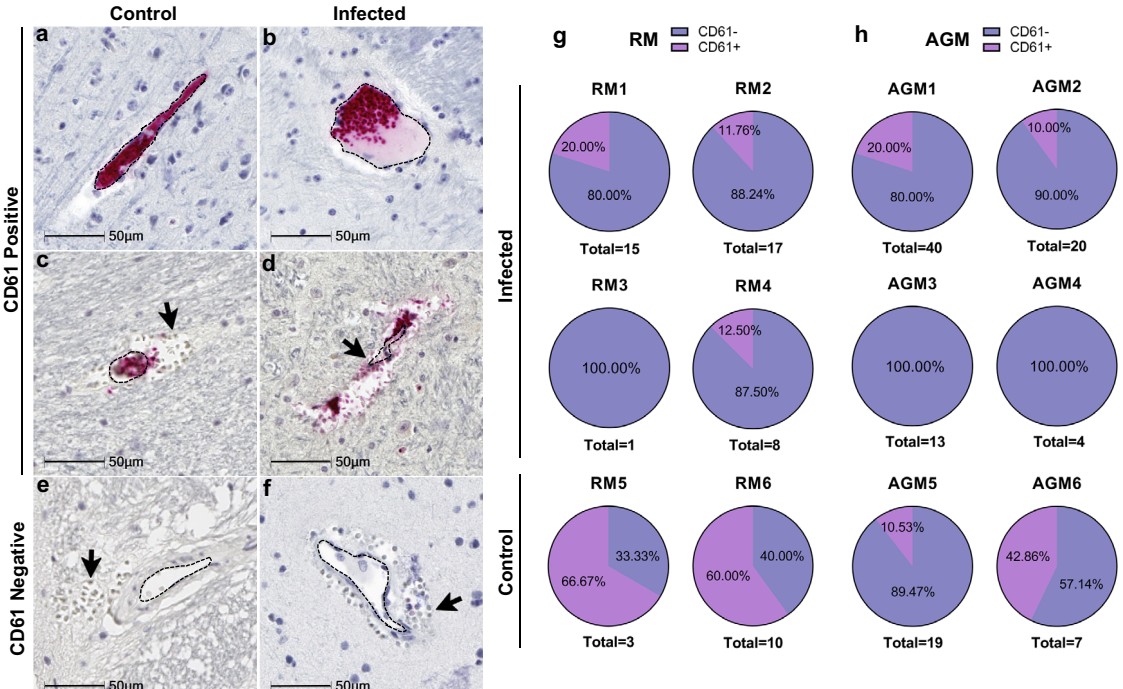

**Fig. 4 Reduced CD61 positive associated-microhemorrhages in SARS-CoV-2 infected NHPs.** Representative images show CD61 positivity in intact vessels in parietal lobe from control animal AGM5 (**a**) and temporal lobe (**b**) from infected AGM4. CD61 positivity was also seen in association with microhemorrhages, as shown in control cerebellum of AGM5 (**c**) and brainstem of infected RM2 (**d**). Microhemorrhages without CD61 positivity were also observed in both non-infected (RM5 brainstem **e**) and infected (RM1 temporal lobe **f**) but was more frequent in infection. The percent frequency of microhemorrhages with and without CD61-associated platelet aggregates is shown for each species (**g** and **h**). Immunohistochemical staining for CD61 was performed twice on all brain regions. Source data are provided as a Source Data file. Abbreviations: AGM African green monkey, RM Rhesus macaque. Scale bars = 50 μm.

be a direct consequence of virus infection, as only limited virus was convincingly detected in brain vasculature and did not appear to involve parenchymal cells. Instead, neuronal injury and death most likely occur as a result of energy failure, which is an early consequence of hypoxic-ischemic events. Multiple microhemorrhages, microinfarcts, and hypoxemia appear to play a role in neuronal injury and death observed in these animals.

Consistent with a hypoxic environment, we detected upregulation/stabilization of HIF-1a in infected animals that localized to the brain vasculature and was significantly greater than mock-infected controls in the deep brain regions assessed. This was observed in all infected animals, regardless of disease severity, suggesting reduced brain oxygen may be a common complication of infection. While the mechanism is not yet elucidated, chronic hypoxemia, as well as an exaggerated and prolonged immune response likely play an important role. Indeed, several inflammatory mediators and growth factors have been reported to stabilize and/or promote expression of HIF-1a, including nitric oxide, interleukin 1b, and tumor necrosis factor-a[28–30].

Interestingly, we did not observe HIF-1a upregulation in cerebellar Purkinje cells in any animal. This may be due to the kinetics of HIF-1a expression, which has been shown in a mouse model of chronic hypoxia to peak in Purkinje cells at 4–5 h and return to normoxic levels after 9–12 h in a continual hypoxic environment[31]. These findings suggest that any potential upregulation and/or stabilization of HIF-1a in Purkinje cells had returned to normal levels by the time the animals were euthanized. It is also likely that degenerate Purkinje cells no longer produce HIF-1a. Our conflicting findings in the brain vasculature may be due to continued exposure of these cells to peripheral factors that promote HIF-1a stabilization and/or expression.

A direct role for the virus in HIF-1a upregulation cannot be ruled out, however, the negligible frequency of SARS-CoV-2 infected cells seen in the CNS compartment argues against the virus being a significant factor. A recent RNAseq analysis, however, found increased HIF-1a mRNA in peripheral blood mononuclear cells (PBMC) acquired from SARS-CoV-2 infected human subjects, as compared to healthy, non-infected controls[32]. Additional in vitro analyses suggested SARS-CoV-2 ORF3a protein induces HIF-1a production in transfected cells, as well as several cytokines upregulated in the context of infection[32]. How this translates to HIF-1a expression in vivo, however, remains unclear.

In agreement with most reports of living subjects and those who died from COVID-19[8], we did not detect virus in CSF and found only minimal virus in the brain that appeared to be limited to the vasculature, suggestive of hematological dissemination of virus to the brain. Infection of pericytes, perivascular macrophages, and/or cells within the brain parenchyma cannot be ruled out but was not convincingly demonstrated in these studies. Instead, virus appeared to be restricted to the endothelium, which is consistent with a previous study of human biopsy tissues that demonstrated the principal receptor for SARS-CoV-2, angiotensin-converting enzyme 2 (ACE2), is expressed by endothelial cells throughout the body, including brain[33,34]. More recently, a large autopsy series out of Mount Sinai demonstrated robust ACE2 expression by brain vasculature in patients who died from SARS-CoV-2 infection[20]. This may suggest a greater vulnerability of the brain to infection in the context of severe disease but was not observed in NHPs that developed ARDS in this study. One autopsy report identified virus in a subset of cranial nerves[22], however, these were not available for investigation. Additionally,

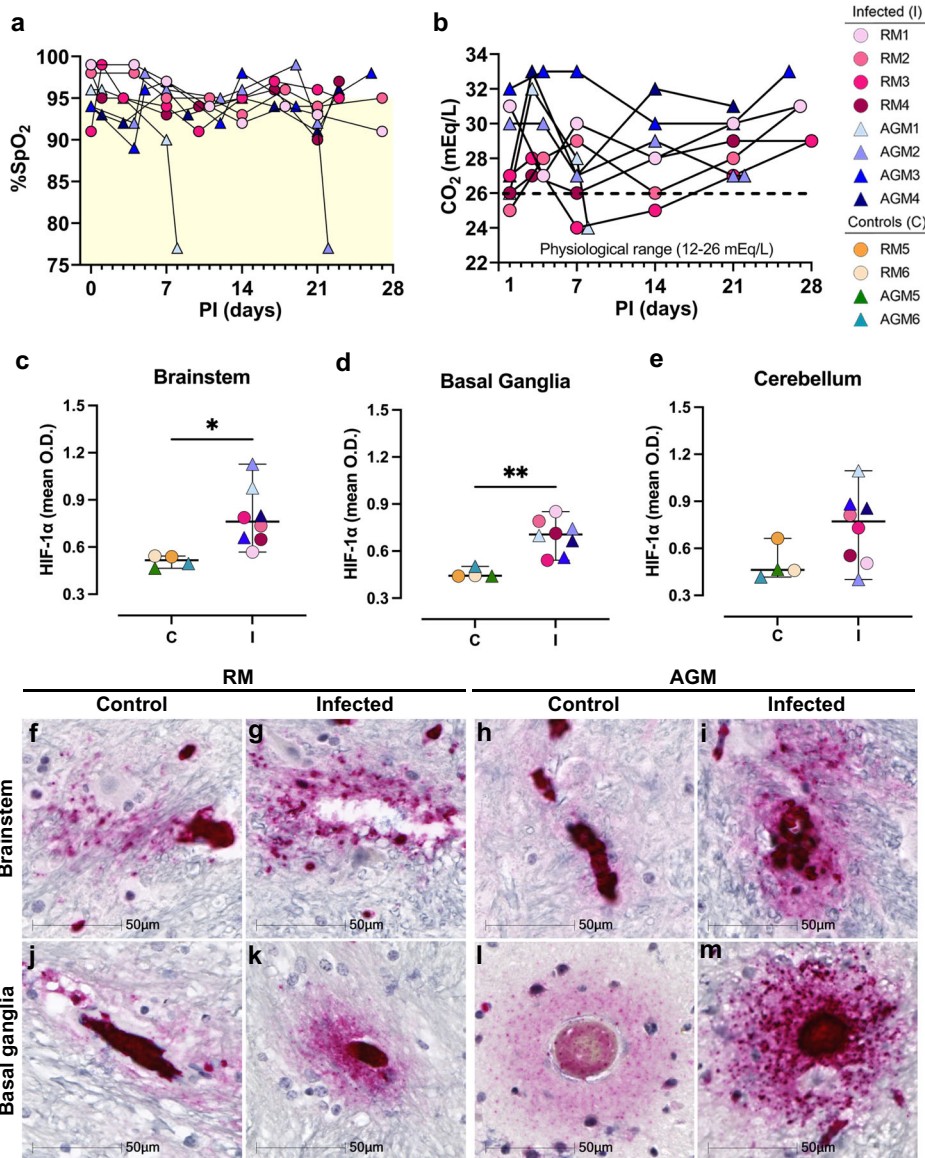

**Fig. 5 Reduced blood oxygen may contribute to brain hypoxia in SARS-CoV-2 infection.** SARS-CoV-2 infection was associated with lower blood oxygen levels (**a**) and increased blood carbon dioxide (**b**). Yellow shading denotes a lower than physiological range of $SpO_2$. HIF-1a expression appeared to be upregulated by cells comprising the vasculature and extended into the parenchyma in the context of infection. Expression was significantly greater than age-matched mock-infected control animals in (**c**) brainstem, *$p = 0.0154$ (95% CI = 0.06550–0.4895) control vs. infected animals and (**d**) basal ganglia, **$p = 0.0016$ (95%CI = 0.1149 to 0.3621) control vs. infected animals. Significant difference was not seen in (**e**) cerebellum ($n = 4$ biologically independent samples in the control group, and $n = 8$ biologically independent samples in the infected group). Statistics were performed with unpaired two-tailed t test, df = 10. Data are expressed as mean ± SEM. When separated by species, statistical significance is retained in the basal ganglia but not brainstem (Supplementary Data Fig. 7). Representative images show low HIF-1α expression in brainstem of mock-infected animals, RM5 (**f**) and AGM5 (**h**), as well as basal ganglia of RM6 (**j**) and AGM5 (**l**). In comparison, HIF-1a is upregulated in brain of infected animals. Representative images include brainstem of RM3 (**g**) and AGM4 (**i**) and basal ganglia of RM3 (**k**) and AGM1 (**m**). Immunohistochemical staining for HIF-1α was performed thrice on the brain regions investigated. Source data are provided as a Source Data file. Abbreviations: PI post-infection. O.D. optical density, %SpO₂ blood oxygen saturation, AGM African green monkey, RM Rhesus macaque, C control, I infected. Scale bars = 50 μm.

the olfactory bulb, which was not recovered from our study animals, may also be an important site for virus entry into the CNS and requires additional investigation.

Notably, the animals in this study were of advanced age, which is associated with a higher risk for the development of cerebrovascular disease among infected patients[8]. Indeed, aging, itself, is the greatest risk factor for cerebrovascular disease, due, at least in part to age-related changes of cerebral vascular structure and/ or function that contribute to reduced cerebral blood flow, which may be further compounded by underlying vascular

pathology[35,36]. This may predispose the aging vasculature to cerebrovascular events, particularly in the context of prolonged systemic inflammation and hypoxemia, which have been shown to contribute to increased vascular permeability through microglia and astrocyte responses[37,38].

Here, we show substantial pathological changes in brain of SARS-CoV-2 infected NHPs that are compatible with autopsy and imaging reports of infected human subjects. Additionally, our pathological investigation suggests a significant role for brain hypoxia in the neuropathogenesis of COVID-19, including

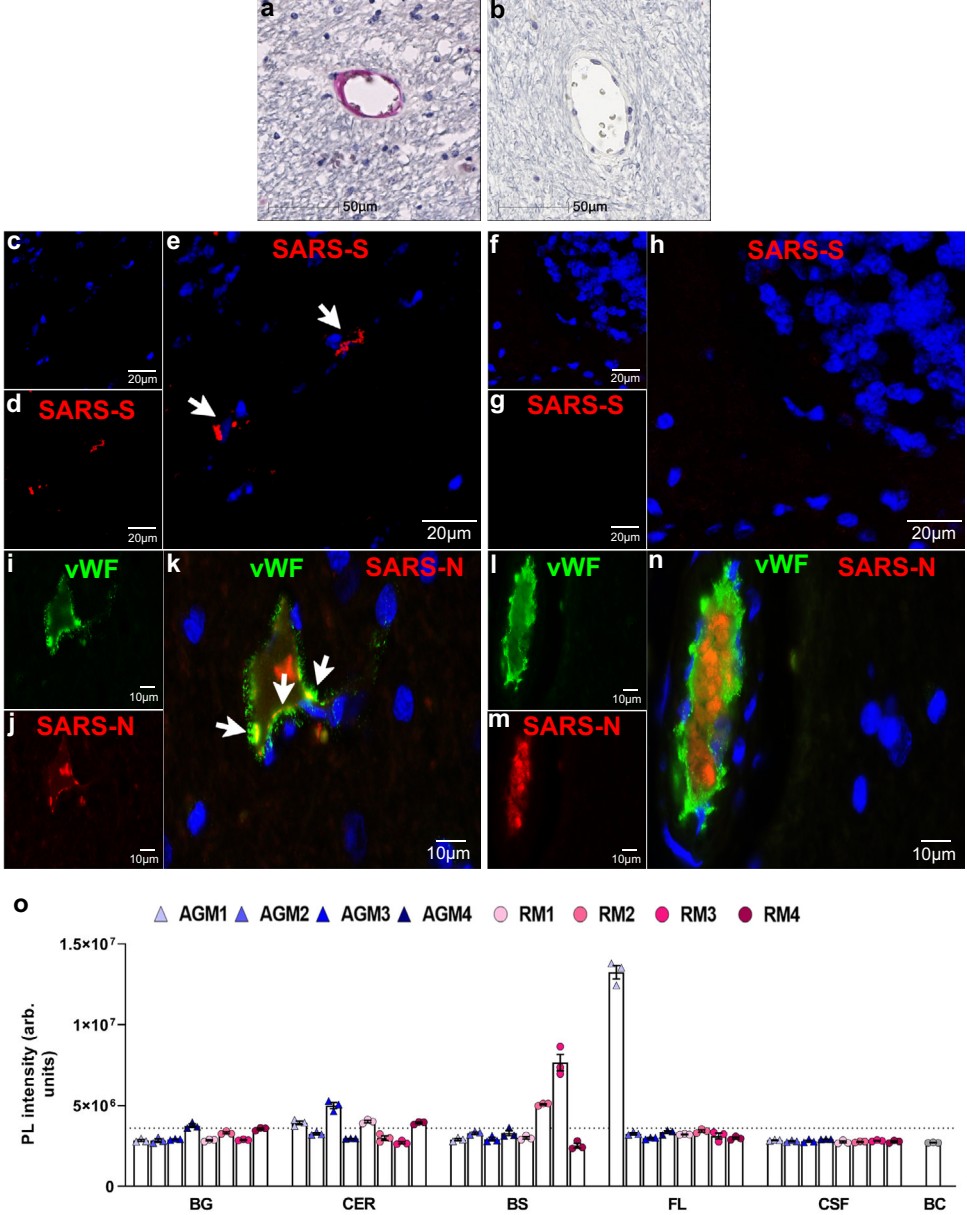

**Fig. 6 SARS-CoV-2 detection in the brain.** Representative single-label IHC shows infrequent SARS-CoV-2 nucleocapsid (SARS-N) positivity in a cerebellar blood vessel of RM1 (**a**). Positivity was not detected in non-infected controls (AGM6 cerebellum b). SARS-CoV-2 spike (SARS-S) mRNA expression was assessed with in situ hybridization (RNAscope) but not seen in control animal tissue (RM6 cerebellum **f-h**). Rare positivity was seen in infection (AGM1 cerebellum **c-e**). RNAscope was performed 7 times on the brain regions investigated. A majority of brain tissue does not show any virus in infected animals which can be seen in the neighboring vessel (RM3 basal ganglia **l-n**) to a vessel with suggestive virus positivity (**i-k**). Endothelial cell infection is suggested by double fluorescent labeling of SARS-N with von Willebrand factor (vWF) (RM3 basal ganglia **i-k**). Merged images show colocalization of SARS-N (red; **j**) with vWF (green; **i**), indicated by white arrows. Blue color represents DAPI labeled cell nuclei. Immunohistochemical staining for SARS-N was performed 12 times on the brain regions investigated. SARS-CoV-2 RNA was detected in the different brain regions of infected animals via a CRISPR-based fluorescent method (**o**) where $n = 3$ repeats of independent samples. Dotted line indicates the cut-off value of positivity equal to $3.6 \times 10^6$ photoluminescence (PL) intensity. Data are expressed as mean ± SEM. Source data are provided as a Source Data file. Abbreviations: BG basal ganglia, CER cerebellum, BS brainstem, FL frontal lobe, CSF cerebrospinal fluid, BC blank control, arb. units arbitrary units, AGM African green monkey, RM Rhesus macaque. Scale bars = 50 μm (**a** and **b**), 20 μm (**c–h**), and 10 μm (**i–n**). Fluorescence images are at 100×.

animals without severe disease. It is reasonable to anticipate that similar findings may occur among human subjects, particularly those with continuing neurological symptoms after recovery from infection[39–41]. For example, an increasing number of retrospective neuroimaging reports have reported cerebral microhemorrhages in critically ill patients with COVID-19[42–44]. Many patients, however, including those who do not require hospitalization, report comparatively milder neurological symptoms that

are not evaluated through neuroimaging. As such, neuropathology among these individuals remains unclear but likely contributes to lingering neurocognitive difficulties reported by a number of convalesced/convalescing patients[45] and warrants further investigation. This further increases the significance of NHPs as a viable model for elucidating the mechanisms that underlie SARS-CoV-2-associated neuropathology that are translatable to human disease, as neuropathogenesis can be more

closely examined in animals that do not experience mortal disease. Additionally, neuropathological complications may contribute to worsening disease among infected patients. For example, damage to the brainstem, which modulates the respiratory cycle by regulating inspiratory and expiratory muscle activity, may contribute to worsening respiratory distress and failure in patients with COVID-19. Additional studies, employing relevant animal models, are warranted and likely to reveal important insight into human disease.

While SARS-CoV-2 neuropathogenic processes are poorly understood, this work reveals infected NHPs are a viable animal model for understanding the neuropathogenesis and potential long-term consequences of infection. We also provide important insight into the mechanisms underlying CNS disease, which was seen even in the absence of severe respiratory disease and may suggest that vascular leakage and hypoxic brain injury is a common complication of SARS-CoV-2 infection and COVID-19. Neuronal degeneration and activation of caspase 3 observed in this study supports this notion and indicates non-reversible neuronal injury may be significant to individuals suffering from PASC. Finally, our findings and conclusions presented herein suggest the need for long-term neurological follow-up of persistently symptomatic convalescent patients.

## Methods

**Ethics and biosafety statement**. All animal studies were approved by the Tulane University Institutional Animal Care and Use Committee and carried out in the Regional Biocontainment Laboratory at the Tulane National Primate Research Center (TNPRC) within an animal biosafety level 3 facility. The TNPRC is fully accredited by the Association for Assessment and Accreditation of Laboratory Animal Care (AAALAC). All animals were cared for in accordance with the Institute for Laboratory Animal Research Guide for the Care and Use of Laboratory Animals, 8th edition. The Tulane University Institutional Biosafety Committee (IBC) approved all procedures for sample handling, inactivation, and removal from BSL3 containment.

**Animal study design**. A total of twelve NHPs, including six Indian-origin RMs (ages 13–21 years) and six AGMs of Caribbean origin (all approximately 16–17 years of age), were included in this study. Four of each species was inoculated with SARS-CoV-2 strain 2019-nCoV/USA-WA1/2020 (MN985325.1) and two of each species were mock-infected with culture media used for virus propagation (Table 1). The viral strain used was isolated from the first confirmed SARS-CoV-2 case in the United States and deposited by the Centers for Disease Control[9]. All animals underwent the same procedures and biological sampling.

All RMs were acquired from the TNPRC specific pathogen-free breeding colony and confirmed negative for simian type D retrovirus (SRV), simian immunodeficiency virus (SIV), simian T cell lymphotropic virus type 1 (STLV1), measles virus (MV), Macacine herpesvirus 1 (MHV1/B virus), and tuberculosis (TB). The AGMs were wild-caught and also confirmed negative for SRV, SIV, STLV, MV, and TB. The AGMs were housed at the Center for over a year before assignment to this study. All animals were tested and found negative for SARS-CoV-2 (antibody and virus) prior to experimental infection.

Two routes of virus exposure, multi-route mucosal and aerosol, were employed to mimic major routes of infection among humans. Two animals from each species were randomly subjected to the different routes of exposure for a total of four animals in each species challenge group. Multi-route exposure included conjunctival, nasal, pharyngeal, and intratracheal routes for a cumulative dose of $3.61 \times 10^6$ PFU (plaque-forming unit). Animals exposed to virus by aerosol received an approximate inhaled dose of $2 \times 10^3$ TCID$_{50}$ (50% tissue culture infectious dose). Study animals were euthanized for necropsy at 24–28 days post infection unless humane endpoints required euthanasia at an earlier time (Table 1). Postmortem examination was performed by a board-certified veterinary pathologist (R.V.B.).

**Quantification of Nasal Swab SARS-CoV-2 subgenomic nucleocapsid mRNA (sg-N mRNA)**. Nasal swab specimens were collected in 200 μL DNA/RNA Shield (Zymo Research) and extracted for viral RNA (vRNA) using the Quick-RNA Viral kit (Zymo Research). Viral RNA Buffer (Zymo) was dispensed directly to the swab in the DNA/RNA Shield (Zymo). A modification to the manufacturers' protocol was to insert the swab directly into the spin column to centrifuge, allowing all the solution to cross the spin column membrane. The vRNA was eluted (45 μL), from which 5 μL was added to a 0.1 mL fast 96-well optical microtiter plate format (Thermo Fisher) for a 20 μL RT-qPCR reaction. The RT-qPCR reaction used TaqPath 1-Step Multiplex Master Mix (Thermo Fisher) along with the following primers and probe: Forward primer: (sgm-N FOR) 5′-CGATCTCTTGTAGAT

CTGTTCTC-3′; Probe: (sgm-N PRB) 5′-FAM TAACCAGAATGGAGAACG CAGTGGG-BHQ1-3′; Reverse primer: (sgm-N REV) 5′-GGTGAACCAAGAC GCAGTAT-3′. The reaction master mix was added using an X-Stream repeating pipette (Eppendorf) to the microtiter plates. Loaded plates were covered with optical film (Thermo Fisher), vortexed, and pulse centrifuged. The RT-qPCR reaction employed the following program: UNG incubation at 25 °C for 2 min, RT incubation at 50 °C for 15 min, and an enzyme activation at 95 °C for 2 min, followed by 40 cycles of denaturation at 95 °C for 3 s and annealing at 60 °C for 30 s. Fluorescence signals were detected with an Applied Biosystems QuantStudio 6 Sequence Detector. Data were captured and analyzed with Sequence Detector Software v1.3 (Applied Biosystems). Equivalent viral copy numbers were calculated by plotting Cq values obtained from unknown (i.e., test) samples against a standard curve representing known viral copy numbers. The limit of detection of the assay was ten copies per reaction volume. A 2019-nCoV positive control (IDTDNA) was analyzed in parallel with every set of test samples to verify the RT-qPCR master mix and reagents were prepared correctly. A non-template control was included in the qPCR to ensure there was no cross-contamination between reactions.

**Immunohistochemistry**. IHC was performed on 5 μm zinc formalin-fixed paraffin-embedded (FFPE) brain sections[46]. Sections were deparaffinized in xylenes and rehydrated through an ethanol series ending in distilled water. Heat-mediated antigen retrieval was carried out in a vacuum oven with Tris-EDTA buffer (10 mM Trizma base, 1 mM EDTA, 0.05% Tween 20, pH 9.0) or sodium citrate buffer (10 mM sodium citrate, 0.05% Tween 20, pH 6.0). All washes were performed using tris buffered saline containing Tween 20 (TTBS; 0.1 M Trizma base, 0.15 M NaCl, 0.1% Tween 20, pH 7.4). Following antigen retrieval, tissues were blocked with 20% normal horse or goat serum. Endogenous biotin was blocked with Avidin-Biotin Solution (Vector Labs). Titrated primary antibodies included anti-cleaved caspase 3 (rabbit polyclonal, 1:250, Abcam, ab2302), anti-von Willebrand factor (vWF, rabbit EPR12010, 1:62.5, Abcam, ab179451), anti-HIF-1a (mouse mgc3, 1:1600, Abcam, ab16066), anti-CD61 (rabbit RM382, 1:125, Invitrogen, MA5-33041), anti-ionized calcium-binding adapter molecule 1 (Iba-1, goat polyclonal, 1:200, Abcam, ab5076), anti-GFAP (rabbit EPR1034Y, 1:500, Abcam, ab68428), anti-HLA-DR (mouse TAL.1B5, 1:400, Novus, NB600989), and anti-SARS-CoV-2 nucleocapsid (rabbit polyclonal, 1:125, Novus, NB100-56576). Tissues were incubated with primary antibody overnight at room temperature and detected using the appropriate biotinylated secondary antibody (1:200, Vector Labs, BA-1100, BA-2000, BA-9500) and alkaline phosphatase-Vector Red according to manufacturer instructions (Vector Labs). Tissues were counterstained with Mayer's hematoxylin and coverslipped.

Double labeling of 5 μm FFPE brain tissue was performed by sequential application of primary antibodies with their corresponding secondary[47]. SARS-CoV-2 nucleocapsid was detected with Alexa Fluor 555 goat anti-rabbit IgG (1:500, Invitrogen, A21428). Von Willebrand factor was detected with Alexa Fluor 488 goat anti-rabbit IgG (1:500, Invitrogen, A11008). Controls consisted of brain tissue incubated in blocking buffer only, tissue incubated with one primary and the corresponding secondary antibody, and tissue incubated with fluorophore-conjugated secondaries only. Tissues were coverslipped with Vectashield® HardSet™ Antifade mount with DAPI (Vector Labs).

**In situ hybridization (RNAscope)**. ISH was carried out on 5 μm FFPE tissues using RNAScope® Multiplex Fluorescent V2 Assay Kit (Advanced Cell Diagnostics), according to the manufacturer's directions. Briefly, sections were deparaffinized in xylenes and dried, followed by incubation with hydrogen peroxide. Heat-mediated antigen retrieval was carried out in a steamer with the provided kit buffer. A hydrophobic barrier was drawn around the tissues before treatment with the kit-provided protease reagent and hybridized with the V-nCoV2019-S probe (Advanced Cell Diagnostics) in a HybEZ oven (Advanced Cell Diagnostics). All washes were performed with the kit wash buffer. Signal amplification was accomplished with three successive AMP solutions and HRP channel (Advanced Cell Diagnostics) and visualized with Opal 570 (1:1000, Akoya Biosciences). Autofluorescence was quenched with TrueVIEW® Autofluorescence Quenching Kit (Vector Labs). Positive and negative control tissues and tissues without probe exposure were included in every run to ensure the specificity of staining and assess background.

**Hematoxylin and eosin**. Deparaffinized and rehydrated slides were taken through Hemalast and hematoxylin, followed by differentiator and bluing solutions. After which, slides were dehydrated in 95% EtOH and stained with eosin. Stained slides were dehydrated, cleared, and coverslipped.

**Luxol fast blue**. Slides were deparaffinized and rehydrated through 95% EtOH, then incubated in warmed 0.1% LFB solution. Afterward, slides were washed, dipped in 0.05% lithium carbonate, differentiated in 70% EtOH, and rinsed. Following a check under microscope, the slides were oxidized in 0.5% periodic acid solution, then immersed in Schiff's reagent before rinsing, dehydration, clearing, and coverslipping.

**FluoroJade C.** Five micrometers FFPE tissues were immersed in 0.06% $KMNO_4$ for 10 min and washed. Tissues were then immersed in 0.0002% FluoroJade C (Histo-Chem) containing 0.1% acetic acid in the dark for 20 min, counterstained with 4′,6-diamidino-2-phenylindole (DAPI), washed, and dried at 60 °C. Cleared tissues were coverslipped with DPX mount (Sigma).

**Imaging and quantitation.** Slides were scanned with the Axio Scan.Z1 digital slide scanner (Zeiss). Brightfield images were acquired using HALO (Indica Labs, v2.3.2089.70 and v3.1.1076.405). Fluorescent images were acquired on a Leica DMi8 automated confocal microscope, model SP8, equipped with a Leica imaging software application suite X model LAS X, software v3.5.7.23225 and an Olympus IX73 inverted microscope with cellSens Dimension 3 software v3.1. Colocalization images were created in Photoshop (Adobe, v21.2.0) by overlaying the same image acquired through the appropriate fluorophore filter. Presented images were subjected to brightness, contrast, and/or darken midtones enhancement in Photoshop, applied to the entire image to reduce background.

Threshold and multiplex analyses were performed with HALO algorithms for non-biased quantitation of proteins of interest, without processing. For active caspase 3 hematoxylin-stained nuclei were used to quantify the number of cells and Vector Red intensity above a rigorous threshold accounted for the cells positive. Quantitation of HIF-1a was performed using an area quantification algorithm for Vector Red intensity. Annotations were drawn to outline blood vessel-associated parenchymal stain based on the algorithm results. The annotated area was analyzed for OD of Vector Red staining. The average OD within the annotated area was calculated in HALO per tissue section.

Microhemorrhages were independently counted and annotated within HALO on seven distinct 5 µm CD61-immunostained regions of the CNS from all infected and control animals by two individuals. Counts were normalized by area of each tissue section. Microhemorrhages were defined by the presence of blood vessels with red blood cell extravasation (>10 red blood cells on the parenchymal side of an unbroken blood vessel). Normalized microhemorrhage counts were plotted for each specific brain region and total regions investigated. $CD61^+$ aggregates within blood vessels were counted and annotated on HALO on seven distinct 5 µm sectioned regions of the CNS from all infected and control animals by two individuals. Blood vessel-associated $CD61^+$ thrombi were defined by aggregated CD61 stained platelets within a vessel.

**RNA isolation from whole tissues.** Dissected frontal lobe, basal ganglia, cerebellum, and brainstem were collected fresh and immediately frozen at necropsy. One milliliter of Trizol LS (Thermo Fisher) was added to 100 mg of thawed tissue and homogenized in gentleMACS M tubes using a gentleMAC Dissociator (Miltenyi Biotec). The resulting lysate was then centrifuged at $3000 \times g$ for 5 min and supernatant transferred into a 2 mL microcentrifuge tube. An equal volume of ethanol (95–100%) was added to the sample in Trizol LS (1:1) and mixed well. The resulting mixture was transferred to a Zymo-Spin III CG Column in a 2 mL collection tube (Zymo) and centrifuged for 30 s. The column was washed with RNA Wash Buffer (Zymo), followed by treatment with DNase I for 30 min to remove residual genomic DNA. The column was washed with RNA Wash Buffer (Zymo) and RNA eluted with 45 µL of DNase/RNase-free water (Thermo Fisher).

**CRISPR-based fluorescent detection system (CRISPR-FDS).** CRISPR-FDS reaction was carried out with the following steps[15]. Isolated RNA samples were mixed with one-step RT-PCR mix containing 2× Platinum™ SuperFi™ RT-PCR Master Mix (Thermo Fisher), forward primer (10 µM), reverse primer (10 µM), SuperScript™ IV RT Mix (Thermo Fisher), and nuclease-free water. Samples were then incubated in a T100 thermocycler (Bio-Rad) using a cDNA synthesis protocol, immediately followed by a DNA amplification protocol. CRISPR-FDS reactions were performed as follows: a sample RT-PCR reaction was transferred to a 96-well half-area plate and mixed with CRISPR reaction mixture containing 10X NEB-uffer™ 2.1, gRNA (300 nM), EnGen® Lba Cas12a (1 µM), fluorescent probe (10 µM), and nuclease-free water. After incubation at 37 °C for 20 min in the dark, fluorescence signal was detected using SpectraMax i3x Multi-Mode Microplate Reader (Molecular Devices). A positive sample was defined as any specimen with a CRISPR-FDS signal that was greater than the cut-off threshold of $3.6 \times 10^6$ photoluminescence (PL) intensity (arb. units).

**Statistics.** Kolmogorov-Smirnov normality test, Mann–Whitney $U$ test, and Student's unpaired two-tailed $t$-tests were performed with GraphPad Prism software, v9.0.2. When separated by species the number of controls was below the detectable limit for the Kolmogorov–Smirnov normality test. Data were defined as gaussian or non-gaussian based on the overall distribution. $P$ values $\leq 0.05$ were considered significant.

**Reporting summary.** Further information on research design is available in the Nature Research Reporting Summary linked to this article.

## Data availability

https://figshare.com/articles/dataset/COVID_NHP_CNS_Source_Data/19241727. https://doi.org/10.6084/m9.figshare.19241727. Source data are provided with this paper.

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

## Acknowledgements

The SARS-CoV-2 sg-N mRNA assay was developed and kindly provided by Dennis Hartigan-O'Connor and Joseph Dutra. This study was supported by the NIH-funded base grant to the TNPRC (P51OD011104). Additional support was provided through Tulane startup funds and Emergent Ventures at the Mercatus Center, George Mason University Fast Grants for COVID-19 to TF; Pilot Grant from P51OD011104 to J.R.; and Weatherhead Presidential Endowment to T.Y.H. G.J.B. was supported by Tulane startup funds.

## Author contributions

J.R., R.B., T.F., R.V.B., L.A.D.M., K.R.L., and C.J.R. designed the study. R.B. and L.A.D.M. provided animal clinical data and interpretation. C.J.M., C.C., R.S., N.G., K.H., K.C., G.L., and N.J.M. participated in tissue acquisition and processing and performed experiments. CRISPR-FDS data acquisition and interpretation were done by B.N., Z.H., and T.Y.H. I.R., M.G.M., L.M.H., and T.F. performed experiments and acquired, analyzed, and interpreted data. Necropsies and sample collection were performed by R.V.B. Manuscript and figures were prepared by T.F., I.R., L.M.H., and M.G.M. and edited by J.R., R.B., G.J.B., R.V.B., and L.A.D.M. All authors have read and approved the manuscript.

## Competing interests

The authors declare no competing interests.
