## [Peer Review File · Nature Communications]

Neuropathology and Virus in Brain of SARS-CoV-2 Infected Non-Human PrimatesEditorial Note: This manuscript has been previously reviewed at another journal that is not operating a transparent peer review scheme. This document only contains reviewer comments and rebuttal letters for versions considered at Nature Communications.

Reviewers' Comments:

Reviewer #2:

Remarks to the Author:

In this revised manuscript, the authors address many of the original reviewer comments. Major issues remaining include confirming the suitability of this model for the study of human SARS-CoV-2 neuropathogenesis, which could be accomplished by a more thorough review of the similarities and differences to the various human neuropathology case series (PMID: 33031735, 32530583, 33015653, 33002281, 32851506, etc.). The authors incorrectly state in the response letter that multiple areas in the human brains were not examined, and that impressions were largely restricted to H&E. Of particular relevance is the impact of inflammation, which has been reported to be a major contributor to brainstem pathology, despite the lack of specificity for SARS-CoV-2. Other issues include: 1) The difficulty of reading white/text and scale bars with the histology images; 2) differences in magnification between images of infected animals and controls (Figure 4 A & B, and Figure 5); 3) difficulty in identifying the cells types infected in Figure 5 based on virus ISH and DAPI alone; 4) ambiguity of cell types staining for virus in Extended Figure 1 (any endothelial cells staining similar to brain or just macrophages or pneumocytes).

Reviewer #3:

Remarks to the Author:

I was specifically asked to review the rebuttal to comments from reviewer #1. The overall comment from reviewer #1 was that data is lacking to show a definitive link between SARS-CoV-2 infection and the observed neuropathology.

While the authors have tried to address several concerns, the main concerns stems from the experimental design which was lacking control animals. This being an NHP study, often historical samples are used, however, given that (mock) inoculation and sampling was not performed in the same way and in the same experiment, it is not a proper control to use. In addition, the findings are very variable and not statistically significant, making it impossible to make any conclusions.

Introduction to the Revised Manuscript

We greatly appreciate the attention the reviewers have afforded our manuscript and the valuable critiques and suggestions for improvement. Indeed, we are not in disagreement with the noted concerns and comments and have worked to appreciably improve the study, which we believe to still be of high significance. Specifically, we have included control animals that underwent the same study protocol as the infected animals, making these more useful for evaluating the impact of infection on neuropathogenesis. We have also included additional immunohistopathological studies to evaluate our model more appropriately against human disease, as well as provide insight into the underlying mechanisms. The manuscript underwent extensive revision and we have revised and added new figures to better describe the neuropathological findings and address the concerns raised. Specifically,

- Figure 1 demonstrates neuroinflammation through Iba-1, HLA-DR, and GFAP immunopositivity.
- Figure 2 is a revision of the neuronal injury and cleaved caspase 3 images, shown in the original submission, and quantitation that demonstrates statistically significant expression of cleaved caspase 3 in infected animals, as compared to age-matched mock-infected control animals. Extended Data Figure 2 shows the regional quantitation of cleaved caspase 3.
- Figure 3 was amended to include microhemorrhages seen in the mock-infected controls and quantitated microhemorrhages in total brain of all animals, which are significantly increased in basal ganglia from infected animals. Extended Data Figure 4 shows the regional quantitation of microhemorrhages.
- Figure 4 reveals new data, showing microhemorrhages with and without CD61 positivity.
- Figure 5 reveals new data, showing HIF-1 α expression in brain, which is significantly upregulated in the context of infection.
- Figure 6 was amended to include SARS spike (SARS-S) RNAscope on brain of infected and mock-infected animals. IHC double label of SARS nucleocapsid (SARS-N) and von Willebrand factor (vWF) or CD31 were also added.
- Extended Data Figure 1 was updated to include mock-infected controls.
- Extended Data Figure 5 is new and demonstrates SARS-S RNAscope in lung and brainstem of infected and mock-infected animals.

Responses to the specific concerns are detailed below.

Reviewers' comments:

Reviewer #2 (Remarks to the Author):

In this revised manuscript, the authors address many of the original reviewer comments. Major issues remaining include confirming the suitability of this model for the study of human SARS-CoV-2 neuropathogenesis, which could be accomplished by a more thorough review of the similarities and differences to the various human neuropathology case series (PMID: 33031735, 32530583, 33015653, 33002281, 32851506, etc.). The authors incorrectly state in the response letter that multiple areas in the human brains were not examined, and that impressions were largely restricted to H&E. Of particular relevance is the impact of inflammation, which has been reported to be a major contributor to brainstem pathology, despite the lack of specificity for SARS-CoV-2.

Response:

We have revised the manuscript to include a more direct comparison with reports of human autopsy series, including similarities and differences. We regret the comment "...impressions were largely restricted to H&E" suggested a lack of scholarly review of the current literature. We realize this was incorrectly written, as we were specifically referring to our findings related to

neuronal injury and apoptosis and not to the inflammatory changes in the brain. Indeed, we do not disagree with the impact of neuroinflammation on CNS injury in the context of infection. This was anticipated and the first thing that we had evaluated in our model through Iba-1, HLA-DR, and GFAP immunohistochemistry. These are seen in our model and reveal morphological changes indicative of reactive astrocytes and microglia. Because the high frequency of microhemorrhages and neuronal injury/apoptosis was more suggestive of significant and potentially, long-lasting injury to the brain, that became the primary focus of the manuscript. We recognize our error and have amended the manuscript accordingly, which we believe better describes the translatability of the model. Evidence of neuroinflammation is Figure 1 in the revised manuscript.

Other issues include:

- 1) The difficulty of reading white/text and scale bars with the histology images.
Response: We have changed the white text and scale bars to make these more visible.
- 2) Differences in magnification between images of infected animals and controls (Figure 4 A & B, and Figure 5).
Response: Matched magnification between infected and mock-infected animals has been provided.
- 3) Difficulty in identifying the cell types infected in Figure 5 based on virus ISH and DAPI alone.
Response: Through IHC (viral nucleocapsid protein) and RNAscope (viral spike mRNA) single label studies, it appears that infection in the brain structures investigated is limited to the endothelium, however, this is very rare. This is also in line with our current understanding of ACE2 expression in the CNS. We have included double label IHC studies, using von Willebrand factor (vWF) and CD31 to identify endothelial cells, which shows colocalization with viral antigen. To further evaluate virus in brain, we included the whole brain CRISPR-FDS analysis. Importantly, we were not influenced by existing data, as this work had been performed in our animals prior to reports of human brain. As these reports emerged, we found our findings were, for the most part, in agreement with possible virus dissemination to the CNS. We recognize, however, that this remains an important question and have acknowledged in the manuscript that virus infection of brain parenchyma cannot be ruled out. This is Figure 6 in the revised manuscript.
- 4) Ambiguity of cell types staining for virus in Extended Figure 1 (any endothelial cells staining similar to brain or just macrophages or pneumocytes).
Response: Representative figures showing the presence of virus in the lung were included only to demonstrate that the animals were SARS-CoV-2 infected and to show differences in lung pathology between control and infected animals. Additional findings in lung are out of the scope of this study but have been noted as pneumocytes in separate reports on these same animals (PMC7695721, PMC7648506). This is Extended Data Figure 1 in the revised manuscript.

Reviewer #3 (Remarks to the Author):

I was specifically asked to review the rebuttal to comments from reviewer #1. The overall comment from reviewer #1 was that data is lacking to show a definitive link between SARS-CoV-2 infection and the observed neuropathology. While the authors have tried to address several concerns, the main concerns stem from the experimental design which was lacking control animals. This being an NHP study, often historical samples are used, however, given that (mock) inoculation and sampling was not performed in the same way and in the same experiment, it is not a proper control to use. In addition, the findings are very variable and not statistically significant, making it impossible to make any conclusions.

Response:

We unreservedly agree with this limitation of the study and at the time of the original work, had to work within NIH-imposed constraints put on the use animals for SARS-CoV-2 infection work to ensure that animals remained available for urgent therapeutic and vaccine development. The deleterious effect of infection on the CNS is recognized as an important complication of infection and we have since been granted the inclusion of age-matched, mock-infected RMs (n=2) and AGMs (n=2) as controls that underwent the same procedures as infected animals.

To ensure rigor and reproducibility, experiments on tissues recovered from control animals were performed with the simultaneous use of tissues from infected animals to reduce the impact of technical variation between runs that could confound comparisons. The inclusion of these technically/experimentally relevant age-matched controls informed the need for additional investigation that provides insight into the mechanisms that may underlie neurological injury in aging. Importantly, our findings provide a conceptual framework for advancing our understanding of the mechanisms of SARS-CoV-2 infection-associated CNS injury and development of therapeutic intervention.

Reviewers' Comments:

Reviewer #2:

Remarks to the Author:

This revised manuscript is significantly improved, including the addition of mock infected controls. Details describing these controls should be added to the Methods, and Extended Data Table 1 should include the route of challenge and necropsy (days PI) for consistency with infected animals.

The HIF-1a immunostaining in Figure 5 shows a bit of an unusual pattern, since HIF-1a is a transcription factor that should have predominantly nuclear with some cytoplasmic localization. It is unclear what cells are staining due to the intensity of the red chromogen, perhaps microglia? Are neurons in the markedly affected areas also positive for HIF-1a (e.g. Purkinje cells)?

The discordance between SARS-CoV-2 nucleocapsid IHC in Figure 6A and immunofluorescence in 6J is puzzling, and is suspicious for non-specific staining. In addition, the dot-like staining of CD31 in 6L raises the possibility that macrophages are being stained, rather than endothelial cells which would be expected to have a smoother membranous staining pattern. The limited dot-like staining for viral nucleocapsid antigen versus more diffuse cytoplasmic staining commonly seen in epithelial cells might make more sense in this context. Overall, these findings are consistent with negligible virus in the brain, consistent with data from human autopsy studies.

Reviewer #3:

Remarks to the Author:

authors have sufficiently addressed this reviewers comments

Introduction to the Revised Manuscript

We would first like to thank the referees for their continued time and effort in evaluating our revised manuscript and appreciate the recognition of the improvements made, as well as the helpful suggestions for improving it further. We have amended the manuscript and revised Figures 5 and 6 to address the noted concerns, as described below. Changes to the manuscript are highlighted in yellow to aid the referees in reviewing the revised version.

Reviewer #2 (Remarks to the Author):

This revised manuscript is significantly improved, including the addition of mock infected controls. Details describing these controls should be added to the Methods, and Extended Data Table 1 should include the route of challenge and necropsy (days PI) for consistency with infected animals.

- **Response:** Thank you for this suggestion. We have amended the Methods and Table 1 accordingly. These changes are highlighted to aid your viewing.

The HIF-1 α immunostaining in Figure 5 shows a bit of an unusual pattern, since HIF-1 α is a transcription factor that should have predominantly nuclear with some cytoplasmic localization. It is unclear what cells are staining due to the intensity of the red chromogen, perhaps microglia? Are neurons in the markedly affected areas also positive for HIF-1 α (e.g., Purkinje cells)?

- **Response:** This is an important concern and we have performed additional IHC studies to address this. First, we retitrated the antibody simultaneously on brain tissue from mock-infected and infected animals. Our original titration was 1:400. This is changed to 1:1600 in this revision. The antibody does titrate and is different in infection and disease, which increases my confidence that the staining that we see is not artifact. While the high intensity seen in the brain vasculature remains, we did not dilute the antibody further, as this resulted in loss of signal by cells with less HIF-1 α expression and/or stabilization. All brain regions of interest were redone for this revision at the 1:1600 dilution. The panel constructed with images from the new titration is seen in Figure 5. The quantitation was also redone, using the new titration, which is also included in Figure 5. Our approach used is included as Extended Data Figures 5 and 6.

We recognize that the pattern of positivity can be difficult to see and, in addition to diluting the antibody concentration, have provided new images at a higher magnification. We have also searched the literature and did an 'image search' using Google to help me understand our data. These investigations revealed a variety of staining patterns, including punctate cytoplasmic staining and highly intense staining in some areas of a specific tissue that obfuscates the nucleus versus cytoplasm and is likely high in both areas. This is seen with other chromogens, such as DAB, as well as immunofluorescence. We are confident the HIF-1 α IHC is real and that the differences noted between infected and mock-infect animals reveal important insight into neuropathogenesis.

Finally, the intense HIF-1 α positivity at the vasculature that extends into the brain parenchyma presumably involves multiple cell types, including perivascular macrophages, endothelial cells, pericytes, astrocytes, and microglia, which would all be in the area of the high intensity. We are continuing to explore this important finding to gain insight into the drivers of HIF-1 α , which may involve other factors, including cytokines and/or growth factors at the blood brain barrier (BBB). We do not see significant differences in the frequency of positive neurons between infected and mock-infected animals. Moreover, we do not see positivity in the Purkinje cells. This may reflect the kinetics of HIF-1 α , which is discussed in the manuscript. These additions/changes to the manuscript are highlighted to aid in the review.

The discordance between SARS-CoV-2 nucleocapsid IHC in Figure 6A and immunofluorescence in 6J is puzzling and is suspicious for non-specific staining. In addition, the dot-like staining of CD31 in 6L raises the possibility that macrophages are being stained, rather than endothelial cells which would be expected to have a smoother membranous staining pattern. The limited dot-like staining for viral nucleocapsid antigen versus more diffuse cytoplasmic staining commonly seen in epithelial cells might make more sense in this context. Overall, these findings are consistent with negligible virus in the brain, consistent with data from human autopsy studies.

- **Response:** We recognize and share concerns related to the appropriate identification of virus in the brain, as well as identifying cells vulnerable to infection within the CNS compartment, should virus be

found. Upon viewing the raw images, we agree that the CD31/SARS-N images are not convincing. We have removed the CD31/SARS-N panels and have replaced it with vWF/SARS-N, as to our understanding this antigen is only found on endothelial cells, with the exception of some reports in tumors. The vWF/SARS-N replacement image shares the smoother membranous staining pattern of SARS-N on one side of the endothelial vasculature consistent with Figure 6A.

As noted, there is negligible virus in the brain in human autopsy studies, which we also see in our animal model. As such, colocalization studies can be challenging, as the majority of our single- and double-label, which we have completed 19 times, do not have detectable virus. We are mindful of the possibility that other cell types may harbor productive virus in brain and have addressed this in the discussion. For this study, the endothelial cell is most convincingly infected, albeit very limited.

Reviewer #3 (Remarks to the Author):

authors have sufficiently addressed this reviewers' comments

Reviewers' Comments:

Reviewer #2:

Remarks to the Author: